# Treatment Strategies in Neutrophilic Dermatoses: A Comprehensive Review

**DOI:** 10.3390/ijms242115622

**Published:** 2023-10-26

**Authors:** Grisell Starita-Fajardo, David Lucena-López, María Asunción Ballester-Martínez, Montserrat Fernández-Guarino, Andrés González-García

**Affiliations:** 1Systemic Autoimmune Diseases Unit, Department of Internal Medicine, Hospital Universitario Ramón y Cajal, IRYCIS, 28034 Madrid, Spain; grisell.starita@salud.madrid.org (G.S.-F.); david.lucena@salud.madrid.org (D.L.-L.); 2Department of Dermatology, Hospital Universitario Ramón y Cajal, IRYCIS, 28034 Madrid, Spain; 3Faculty of Medicine and Health Sciences, Universidad de Alcalá (UAH), 28801 Alcalá de Henares, Spain

**Keywords:** dermatosis, autoinflammatory, immunomodulation, therapy

## Abstract

Neutrophilic dermatoses (NDs) are a group of noninfectious disorders characterized by the presence of a sterile neutrophilic infiltrate without vasculitis histopathology. Their physiopathology is not fully understood. The association between neutrophilic dermatoses and autoinflammatory diseases has led some authors to propose that both are part of the same spectrum of diseases. The classification of NDs depends on clinical and histopathological features. This review focuses on the recent developments of treatments in these pathologies.

## 1. Introduction

The term neutrophilic dermatoses (NDs) was used for the first time in 1964 by Robert Douglas Sweet to describe a case of febrile neutrophilic dermatosis, currently known as Sweet’s syndrome.

The most representative entities within this group include pyoderma gangrenosum (PG), Sweet’s syndrome (SS), pustular dermatoses predominantly subcorneal, generalized pustular psoriasis (GPP), and those secondary to inflammatory bowel disease (IBD) [1].

In this review, the location of the neutrophil infiltrate (epidermis, dermis, and/or subcutaneous, the clinical presentation, and the chronicity, which make them unique from the rest, are used to classify each entity [2]. The first classification of ND was defined by Wallach and Vignon-Pennamen [3]. Given its initial complexity, it was reformulated, prioritizing the location of the neutrophilic infiltrate [4].

The pathogenesis of ND is not clearly understood and has been related to abnormal neutrophil function, inflammasome activation, the malignant transformation of neutrophils that infiltrate the dermis, as well as genetic predisposition [5].

There are a wide variety of inflammatory markers and cytokines expressed, among which are CD3 and CD163, IL-1 (α and β), IFN-γ, IL-2, IL-6, IL-8, IL-17, myeloperoxidase, and TNF-α [6].

A unique case is represented by GPP, whose etiology seems to depend on a loss of function in IL36RN, the gene that encodes IL-36Ra. The secretion of IL-36 by keratinocytes or inflammatory cells and the stimulation of autocrine and paracrine pathways trigger an inflammatory response mediated by cytokines that play a key role in the development of this disease: CXCL1, CXCL2, CCL20, IL-8, IL-12, IL-1b, IL-23, IL-6, and TNF-a.

Subsequently, because of the release of these cytokines, the activation of T cells occurs, which produces secretions of IL-22, IL-17, and IFN-c [7].

The treatment of ND varies according to each subtype. The treatment is based in controlling the underlying disease, if found, using glucocorticoids. In addition, a wide variety of alternatives, such as immunosuppressive agents and antimetabolites, as well as azathioprine, cyclosporine, mycophenolate, and cyclophosphamide, could be added in the case of an absence of response. Biological treatment varies according to the desired therapeutic target, as anti-TNF-α, as well as anti-IL 12/IL-23, anti-IL-17, anti-IL-1, anti-IL- 1β, and anti-IL-36, among others, can be used [1].

The objective of this article is to summarize the therapeutic novelties of the main syndromes that include neutrophilic dermatoses.

## 2. Sweet’s Syndrome

### 2.1. Introduction

As previously mentioned, Sweet’s syndrome was first described in 1964 by Dr. Robert Douglas Sweet, who used the term “acute febrile neutrophilic dermatosis” to describe eight cases of women with systemic symptoms including fever, leukocytosis, and painful plaques with an extensive neutrophilic infiltrate on the histopathology. In all cases, infections had been ruled out; all of them also exhibited a good response to treatment with steroids [1,5].

SS is characterized by erythematous papules or plaques that can occur on the trunk, upper extremities, head, and neck. It is characteristically accompanied by fever and neutrophilia. It predominantly affects the female sex and can occur at any age. SS could be idiopathic or secondary to other pathologies, as well as infections, inflammatory bowel disease, endocrinopathies, autoimmune diseases, and tumors, among others. It could also occur with drugs, such as antibiotics, retinoids, antiepileptics, and anti-TNF-α.

Histologically, it presents with a dense neutrophilic infiltrate in the superficial dermis, which may affect the subcutaneous portion.

### 2.2. Pathogenesis

The pathogenesis is not exactly known; currently, the postulated hypothesis is that an external or internal agent triggers an activating signal with the consequent release of proinflammatory cytokines, among which are IL-1, IL-6, and IL-8, causing neutrophilic migration to the dermal region [1,2,3].

In the case of SS, it is crucial to influence the underlying pathology or disorder as the possible causative agent.

### 2.3. Treatment

#### 2.3.1. Topical Therapy

In localized lesions, topical or intralesional glucocorticoids could be used.

#### 2.3.2. Systemic Treatment

In addition, targeted therapy based on glucocorticoids is usually necessary, such as the use of prednisolone, at a dose of 0.5 to 1 mg/kg/day, with a progressive decrease over 4–6 weeks [2]. Our approach is to avoid higher doses of glucocorticoids, and to try to combine another immunosuppressant whenever the disease needs to be controlled. In this scenario, second-line treatments, such as dapsone (100–200 mg/24 h) and colchicine (1.5 mg/24 h), are useful in monotherapy and in combination, with the aim being to decrease the burden of chronic treatment with glucocorticosteroids [1,2]. The evidence regarding biological treatments in SS is limited, and mostly represented by case reports and a few open studies. Likewise, a case of rheumatoid arthritis with SS refractory to initial treatment with prednisone, infliximab, rituximab, methotrexate, abatacept, tocilizumab, and golimumab is described in the literature, which finally responded to therapy with baricitinib [8].

In addition, there is one reported case of a 12-year-old girl diagnosed with CANDLE syndrome (recurrent fever, visceral inflammation, lipodystrophy, and fixed skin lesions) who presented a complete response to tofacitinib [9]. Both cases lead us to consider the possible role of JAK inhibitors in SS.

A special case is SS associated with inflammatory bowel disease, where TNF-α inhibitors, including etanercept, infliximab, and adalimumab, have demonstrated efficacy. Likewise, the role of ustekinumab in Crohn’s disease has been shown, leading to remission in a patient with secondary SS [10,11,12,13]. See Table 1. 

Refractory cases usually require high doses of methylprednisolone (250 mg/24 h for a period of 3–5 days). Rituximab and anakinra are also postulated as favorable alternatives [2,14,15,16,17].

## 3. Pyoderma Gangrenosum

### 3.1. Introduction

This entity was described in 1916 by Brocq, and was later renamed in 1930 by Brunsting, Goeckerman, and O’Leary [18,19]. The prevalence is 5.8 cases per 100,000 inhabitants, predominantly in the female sex.

Pyoderma gangrenosum is frequently shown in association with systemic pathology, including inflammatory bowel disease, rheumatoid arthritis, and hematological malignancies. PG is closely related to exposure to drugs, among which colony-stimulating factors, levamisole, cocaine, and immunomodulatory agents stand out [1,2,3].

There are different types of PG; approximately 85% are classified as classic ulcerative, while the other 15% include bullous, vegetative, pustular, peristomal, and superficial granulomatous variants.

The most common is the ulcerative subtype, which classically presents as single or multiple lesions. It could be a nodule or pustule that rapidly expands to form a painful purplish ulcer with a fibrinous bottom and ulcerative edge. It can occur anywhere in the body and presents pathergy [1,2,3,5].

It is important to emphasize the increase in morbidity and mortality of patients with PG compared with controls matched by age and sex [20,21].

### 3.2. Pathogenesis

The etiopathogenesis underlying this entity is unknown. It is believed to be based on a dysregulation of innate and acquired immunity. There is an alteration in the response of the immune system to superantigens as well as neutrophil, TNF-α, IL-12/IL-23 dysfunction, and genetic predisposition. The role played by neutrophils in this entity is conditioned by defects in chemotaxis, phagocytosis, and bactericidal ability [1,22].

Other mechanisms implied in the etiology of PG have been described. Recent studies have focused on the role of lymphocytes and biomarkers. A clonal expression of T lymphocytes has been observed in intralesional and peripheral blood, mainly for CD3+ and CD163 macrophages [22]. In addition, the overexpression of IL-8, IL-16, IL-18, IL-17, IL-23, MMP (metalloproteinases) 2, 9, and TNF-α, and deregulation between effector T-reg and Th17 cells, has also been demonstrated.

Although its role is not well defined, the expression of IL-1β in PG lesions suggests an autoinflammatory basis with consequent activation of the inflammasome. Additionally, genetics is an essential factor contributing to syndromes that associate PG and autoinflammatory phenomena, as has been verified with mutation in the Proline-Serine-Threonine Phosphatase Interactive Protein (PSTPIP)-1 gene that encodes for the CD2-binding protein 1, located on chromosome 15q, which is involved in PG and associated autoinflammatory syndromes such as PAPA syndrome (pyogenic sterile arthritis, PG, and cystic acne) and PASH syndrome (PG, acne, and suppurative hidradenitis) [22,23].

### 3.3. Treatment

#### 3.3.1. Topical Treatment

Regarding treatment, the first line is based on topical treatment, with both analgesia and topical corticosteroids or tracrolimus [24].

#### 3.3.2. Systemic Treatment

##### Glucocorticoids

Prednisone is usually administered at a dose of 0.5–1 mg/kg/day up to 2 mg/kg/day, with a gradual decrease between 4 and 6 weeks, and can be prolonged until completion at 4–6 months. Some authors recommend methylprednisolone pulses (250–1000 mg) for 3–5 consecutive days with consequent transition to the oral route, thus achieving a faster effect.

##### Immunosuppressants

The commencement of corticosteroid-sparing drugs should be performed as quickly as possible, because high doses of prednisone are associated with high mortality rates [25]. Among them, oral cyclosporine has emerged as the main immunosuppressant at doses of 2–3 to 4–5 mg/kg/day, and could be used both alone and in combination with glucocorticoids. The choice of using cyclosporine against corticosteroids depends on the patient’s comorbidities. Pre-existing conditions that favor the use of corticosteroids over cyclosporine include renal failure or oncology history. However, patients with obesity, diabetes mellitus, osteoporosis, or peptic ulcer will benefit from cyclosporine [26]. In the same way, during the reduction in corticosteroids, dapsone 50–200 mg, azathioprine 100–300 mg/day, methotrexate 10–30 mg/week, cyclophosphamide 1.5–3.0 mg/kg/day, mycophenolate mofetil 2–3 g/day, mercaptopurine, melphalan, thalidomide, and IVIG could be used. In addition, there are case series described in patients with PG refractory to conventional treatment who responded to the association of glucocorticoids with tracrolimus [27,28].

##### TNF-α Inhibitors

Infliximab, adalimumab, etanercept, certolizumab, and golimumab have recognized efficacy in refractory PG and are associated with IBD [26], according to a semi-systematic review published by Ben Abdallah et al. After analyzing 222 articles that included 356 patients with PG treated with TNF-α inhibitors, an 87% response rate and 67% complete remission were described, without finding significant differences between different drugs in terms of effectiveness [23].


**Infliximab**


In a double-blind randomized clinical trial, Brooklyn et al. compared infliximab with a placebo, for which 30 patients with PG and IBD were recruited. Subjects were randomized to receive infliximab 5 mg/kg for 2 weeks. After two weeks of treatment, a response rate of 46% was observed in the infliximab group vs. 6% in the placebo group. Subsequently, given the obtained results, an open label trial was carried out that included the rest of the untreated patients; it was observed that at week 6, 90% had presented remission, with complete resolution in 21% of the patients treated with infliximab [29]. In the review published by Pascal Juillerat et al., wherein 22 articles with 85 patients with a diagnosis of PG associated with IBD being treated with infliximab were analyzed, the efficacy of the treatment in cases of corticosteroid resistance and ulcerative phenotype was concluded [30]. Arguelles-Arias et al. also carried out a retrospective analysis of a cohort of subjects with PG, in which they showed response rates of 92% and 100% with infliximab and adalimumab, respectively [31]. In total, 58 cases of PG without IBD treated with anti-TNF-α have been reported in the literature, of which 41% were idiopathic PG, 17% were associated with rheumatoid arthritis, 10% were post-surgical/traumatic, 7% were associated with hidradenitis suppurativa, 4% were associated with monoclonal gammopathy, and 3.5% were related to cocaine use (levimasol) [32]. There is robust evidence supporting the use of infliximab in PG, both associated with and unrelated to IBD [29,30,31,32]. See Table 2.


**Adalimumab**


The use of adalimumab has been shown to be effective in cases of recalcitrant PG, as demonstrated in some cases reported in the literature [33,34]. Similarly, the case of a patient diagnosed with rheumatoid arthritis treated with methotrexate and etanercept, who developed refractory PG after 2 months of orthopedic surgery, has recently been published [35]. Initially, it was decided to add prednisone, despite which remission was not achieved; thus, it was decided to change etanercept to adalimumab, achieving practical resolution of the lesions after 25 weeks of treatment.

As previously mentioned, Rousset L. et al. described 14 cases of patients with PG not associated with IBD treated with adalimumab, of whom 64% presented complete remission—figures similar to those obtained after treatment with infliximab [32]. Herberger et al. analyzed 52 patients with PG who received treatment with biologics or intravenous immunoglobulins; it was observed that up to 57.1% (16/28) of the patients treated with adalimumab experienced complete remission or improvement of the lesions [38]. Lastly, the use of adalimumab has also been described in autoinflammatory pathologies such as PASH and PAPA syndrome, with the adequate response and resolution of PG [49,50].


**Etanercept**


Etanercept represents a useful option in cases of recalcitrant PG. According to the series of cases collected by Rousset L. et al., 47% (9/19) of patients presented complete remission after treatment with etanercept [32]. Similarly, Herberger et al. conducted a retrospective cohort study in patients with a diagnosis of PG not associated with IBD, in which clinical improvement was evidenced in 71.4% (5/7) of patients treated with etanercept [38].


**Golimumab**


The evidence available in the literature regarding the use of golimumab in PG refers to a few isolated cases. Diotavelli F. et al. published the case of a 68-year-old subject with a history of ulcerative colitis, who received treatment with infliximab and adalimumab, and then suffered an episode of rectocolitis and the appearance of an ulcer in the distal third of the left leg. After performing a biopsy and screening for infectious disease, treatment with glucocorticoids and golimumab was started, with excellent clinical evolution [37].


**Certolizumab**


The use of certolizumab in PG is limited to specific cases, as we previously observed in the series of cases reported by Rousset L. et al., which added a subject who received treatment with certolizumab with an adequate response and disappearance of lesions [32] Similarly, a case of a patient with Crohn’s disease with significant skin involvement due to disseminated PG, refractory to multiple lines of treatment, was reported; they received certolizumab with systemic glucocorticoids and tacrolimus, obtaining response and resolution of lesions after 11 months of treatment [36].

##### IL-1 Inhibitors


**Anakinra**


Anakinra acts by inhibiting IL-1 α and β; it plays a fundamental role in the treatment of monogenic autoinflammatory syndromes, such as CAPS (cryopyrin-associated periodic syndromes) and Still’s disease in adults [51]. If we delve into the pathophysiology of PG, the expression of IL-1β and its receptor has been observed in PG lesions, hence its use in this pathology [1,22,26]. Based on these data, anakinra could play a role in recalcitrant PG and autoinflammatory syndromes, such as PASH and PAPA syndrome, with adequate clinical responses to treatment [45,46,47,48,52]. The dose used was 100 mg daily subcutaneously, in combination or not with glucocorticoids, which can be increased to 200 mg daily (2 mg/kg day) in the case of an inadequate clinical response.


**Canakinumab**


The use of canakimumab is based on the inhibition of IL-1β, widely used in CAPS and adult-onset Still’s disease [51]. Regarding the evidence supporting its use, a clinical trial published by Kolios A.G. et al. presented the selection of five subjects diagnosed with steroid-refractory PG. The intervention consisted of receiving 150 mg of canakimumab subcutaneously at week 0, with a possible extra dose after two weeks in case of refractoriness, as well as an optional dose of 150–300 mg at week 8 based on clinical assessment. At week 16, 80% of the patients presented improvements and 60% exhibited complete remission [44].

##### IL-17A Inhibitors


**Secukinumab**


There are no clinical trials or case series that support the use of secukinumab; therefore, its evidence is relegated to reports of isolated clinical cases. The PG subtype that has presented a favorable response in the reported cases is ulcerative, one of which is post-surgical. Usually, they are patients who have presented refractoriness to conventional treatment lines, in which it is used as an off-label drug. The dose used in both clinical cases was 300 mg subcutaneously initially, and later at weeks 0, 1, 2, 3, and 4, followed by a monthly maintenance dose [42,43].

##### IL-23 Inhibitors


**Ustekinumab**


Ustekinumab is used in the management of PG, including severe and recalcitrant cases, at a dose of 90 mg every 8 weeks or at week 0, week 4, week 8, and every following 8 weeks for a minimum of 12 months. It is usually administered in the absence of concomitant therapy [38,39,40,41]. It could be an option than may control both PG and Crohn disease. As we have seen, it is an interesting option in cases of intolerance, contraindication, or resistance to TNF-α inhibitors or steroids [53].

##### Anti-IL-17 Receptor


**Brodalumab**


Two cases are described, both with coexistent hidradenitis suppurativa, in which a weekly subcutaneous dose of 210 mg/1.5 mg was used as HS treatment, achieving practical resolution of the PG lesions [54].

##### Anti-IL-6 Receptor


**Tocilizumab**


As we already know, IL-6 plays a fundamental role in the inflammatory cascade; hence, its blocking can have favorable effects in multiple pathologies dependent on its activation. Two cases of favorable response to tocilizumab are described in the literature. The first describes a patient with rheumatoid arthritis and interstitial lung disease who was diagnosed with PG. Initially, treatment with glucocorticoids was established at a dose of 0.5 mg/kg/day with poor response. Subsequently, given the contraindication for the use of anti-TNF due to the pulmonary involvement, the use of tocilizumab was chosen, with an initial dose of 162 mg subcutaneously every two weeks. Finally, the patient presented adequate clinical evolution after IL-6 blockade with practical resolution of the lesions [55]. The second case was a patient with a longstanding history of PG refractory to multiple lines of treatment who was recently diagnosed with Takayasu’s arteritis. Given this situation, the decision was made to start tocilizumab, with a favorable response: control of the disease and healing of lesions [56].

##### Anti-IL-23


**Tildrakizumab**


There are three cases in the literature in which tildrakizumab has been used as a treatment for PG. The cases described in these reports were of ulcerative PG and vegetative PG subtype. The patients had coexisting entities such as polymyalgia rheumatica and PASH. The initial dose used was 100 mg subcutaneously in weeks 0 and 4, and then later every 12 weeks; after the start of the treatment, it was possible to reduce the concomitant immunosuppressive treatment as well as heal the lesions [57,58,59].

## 4. Hidradenitis Suppurativa

### 4.1. Introduction

Hidradenitis suppurativa (HS) was initially described in 1839 by the French surgeon Verneuil [60]. It is a chronic, recurrent, and debilitating inflammatory disease that usually presents after puberty with deep, inflamed, and painful lesions, spreading exclusively to body areas with the presence of apocrine glands, the most affected regions being the axillary, anogenital, and inguinal regions [61].

Regarding its epidemiology, despite the fact that the real prevalence is unknown, it is estimated that it can be between 0.00033% and 4.10%, with a predominance in the female sex and the African American race. There are no exact data on the actual incidence, although according to a retrospective study it could be 11.4 cases per 100,000 inhabitants, with twice the number of cases in women. The age distribution is situated between 18 and 44 years.

Multiple comorbidities are associated with HS, including obesity and smoking. In addition, there is a higher prevalence of psoriasis among patients suffering from this entity. Approximately 40% of HS patients have an affected family member, implying a genetic predisposition.

### 4.2. Pathogenesis

In relation to its etiopathogenesis, the primary event consists of follicular hyperkeratosis that, consequently, produces a rupture of the hair follicle and inflammation of the apocrine glands. Both interleukin IL-17 and TNF-α play fundamental roles. Elevated levels of TNF-α in the skin and serum IL-17 correlate with the severity of the disease. The involvement of sex hormones in this pathology is not exactly known [62].

### 4.3. Classification

Treatment depends on the severity of the disease, which is graded according to the Hurley scale:Hurley I—Abscesses, single or multiple, without fistulous tracts or scarring;Hurley II—Abscesses separated from each other and recurrent with fistulous tracts and scarring;Hurley III—Multiple abscesses with fistulous tracts and abundant scarring [63].

### 4.4. Treatment

#### 4.4.1. Non-Pharmacological Treatments

Therapeutic options vary from pharmacological therapy to surgical interventions. Among non-pharmacological interventions, lifestyle changes stand out. Cessation of smoking seems to have potentiating effects in terms of reducing the severity of the disease [64]. In the systematic review carried out by Weber et al., which included a total of 2829 patients, a significant but weak improvement was observed in the patients who lost weight and changed their diet. The same occurred with those who were supplemented with oral vitamin D and zinc [65].

#### 4.4.2. Topical Treatment

Topical therapy is the first-line therapy in localized forms. Its use is also recognized as a complement to systemic therapy in more complex forms. It consists of the use of antiseptics, topical antibiotics, keratolytics, and/or intralesional corticosteroids.

#### 4.4.3. Systemic Treatment

##### Antibiotics

Regarding systemic treatment, we must emphasize the role of antibiotics. Among them, in the first line, we find oral tetracyclines, which are very useful because of their anti-inflammatory effects. Subsequently, in the second and third lines, we find clindamycin–rifampicin and metronidazole/moxifloxacin/rifampicin, respectively. There is also a report of the use of ertapenem in cases refractory to other lines of antibiotic therapy, with an adequate response.

The use of dapsone should be relegated to third-line treatments, especially in those patients with moderate involvement (Hurley I–II) if the first- and second-line treatments have failed. The dose used varies between 25 and 200 mg daily, and it is recommended for at least 3 months [66].

There is little evidence about the use of zinc in patients with Hurley stage I and II; despite this, it seems that its key role in innate immunity could be favorable in certain cases [67].

##### Retinoids

Retinoids are also a fundamental part of the therapeutic arsenal for this pathology. They are usually relegated to the second or third line, when antibiotics have failed. The best known is acitretin, at doses of 0.2 to 0.88 mg/kg per day. It seems that there are many factors that predispose one to a better response with this drug, among which are a family history of HS, elevated levels of activity, and a history of acne conglobata [68,69,70]. In the case of isotretinoin, evidence about its use is contradictory; usually, the dose used is 0.5 to 1.2 mg/kg [69,70].

##### Biological Treatment

In recent years, the use of biological therapy has increased. Immunomodulation is essential in refractory or severe cases. Treatments are based on IL-1 (anakinra), IL-12/23 (ustekinumab), IL-17 (secukinumab), and the TNF-alpha (infliximab, adalimumab) blockade. To date, only adalimumab has been approved for first-line therapy, and infliximab for the second line [66].

TNF-α inhibitors


**Adalimumab**


Adalimumab is a recombinant human monoclonal antibody against TNF-α effective for moderate and severe cases. The most relevant clinical trials in relation to its effectiveness are PIONEER I and II [71], which included a total of 633 patients who were randomized to receive placebo or adalimumab. The design was similar in both trials, the difference being that in the second trial, concomitant treatment with oral tetracyclines was allowed. The primary endpoint was clinical response, defined as a greater than 50% reduction in lesions, this being significantly greater in adalimumab-treated patients compared with placebo (41.8 vs. 26% in PIONEER I and 58.9 vs. 27.6% in PIONEER II). Based on these results, adalimumab was approved by the FDA for moderate to severe HS.


**Infliximab**


Infliximab is a monoclonal chimeric antibody against TNF-α, indicated as the second-line treatment in moderate–severe cases refractory to adalimumab. Its use in HS has not been specifically approved in this entity; despite this, the European guidelines recommend a dose of 5 mg/kg body weight administered on weeks 0, 2 and 6, and then regularly every 8 weeks [68].

IL-1 inhibitors


**Anakinra**


Anakinra is a recombinant human IL-1 receptor antagonist that blocks the inflammatory effects of IL-1. Its efficacy has been demonstrated in two clinical trials. The first was a randomized clinical trial in 20 patients with Hurley stage II/II, in which the group randomized to receive anakinra presented clinically significant responses of 78%, while in the placebo group, this was 30% [72]. Similarly, Leslie et al. carried out an open clinical trial with five patients diagnosed with HS in the moderate–severe phase who received treatment with anakinra for 16 weeks, with an objective decrease in activity. The dose usually administered is 100 mg/day subcutaneously [73]. The HS ALLIANCE group currently recommends it as a third-line therapy in cases of the failure of TNF-alpha inhibitors [74].

IL-23 inhibitors


**Ustekinumab**


Ustekinumab is a human IgG1 class monoclonal antibody that modulates IL-12 and IL-23 signaling. Regarding evidence of its use, a cohort of 17 patients with Hurley stage II–III HS received 45 mg s.c. at weeks 0, 4, 16, and 28. Of the subjects included, 82% presented a great clinical improvement [75]. The North American and European guidelines consider it in cases of refractoriness to previous lines [68,76].

L-23 inhibitors


**Secukinumab**


Secukinumab is a human monoclonal antibody directed against IL-17A. There seems to be an increase in IL-17A in the blood of patients with HS, related to the severity of the disease. Current clinical guidelines have not mentioned the use of secukinumab in this entity. However, there are several trials that have reported its use [77,78], including SUNSHINE and SUNRISE, phase 3 multicenter randomized clinical trials with a total of 541 and 543 patients, respectively. The included cases were patients with moderate–severe HS who were randomized to receive secukinumab 300 mg s.c. every 2 weeks or every 4 weeks, or a placebo. The primary endpoint was clinically significant improvement, defined as a more than 50% improvement in lesions. Of the patients randomized to secukinumab, it was verified in both trials that the most effective regimen was every 2 weeks. Despite this, it is necessary to continue investigating the indication in this entity [78]. See Table 3. 

Regarding immunosuppressive treatment, the following drugs are distinguished.

#### 4.4.4. Glucocorticoids

The European, North American, and HS ALLIANCE clinical guidelines indicate glucocorticoids as a treatment in severe cases, or in order to perform bridging therapy with another drug [68,74,76]. The North American and European guidelines recommend a dose of 0.5–0.7 mg/kg oral prednisolone. Prolonged treatments are not recommended due to their potential side effects [68,76].

#### 4.4.5. Immunosuppressants


**Cyclosporin A**


Evidence about the effectiveness of cyclosporin Ais scarce [79,80]. The clinical guidelines recommend cyclosporine in cases of first-, second-, or third-line failure. The doses usually used are 2–6 mg/kg once daily for various durations (5 to 30 weeks) [66,68,79,80].

#### 4.4.6. Hormone Therapy

Hormone therapy is one option to consider; androgens promote the occlusion of the hair follicle through a proliferation of keratinocytes, giving rise to acanthosis and keratosis follicularis. The predominance in females—changes with menstruation, a worsening in the menopause, and improvements during pregnancy—make this theory plausible [78,81].

Among other treatments, we find metformin, which could be beneficial in patients with polycystic ovaries and diabetes mellitus. The North American guidelines recommend a dose of 500 mg two or three times a day [69,76,82].

Finasteride can also be considered; it could be useful via its action of inhibiting the androgen-mediated exacerbation of HS. The North American guidelines recommend doses of 1.25 to 5 mg/d, which have been reported as effective in several trials. It can be used both in monotherapy in moderate–severe cases and in additional therapy in severe cases [76,83].

#### 4.4.7. Other Therapies

Finally, it is worth mentioning spironolactone. According to the North American guidelines, patients who used spironolactone at doses of 100–150 mg daily showed clinical improvements after 3–6 months of treatment [76].

Among experimental therapies, the role of the botulinum toxin should be highlighted. Although the underlying mechanism is not clear, it seems to be a plausible option for patients with extensive Hurley III involvement [84,85].

The surgical option should be considered in patients with severe involvement with chronic lesions that do not respond to conventional therapy. Amongst the different surgical options, we find radical surgical excision, deroofing, drainage, carbon dioxide laser therapy, and YAG laser therapy [64,74,76].

## 5. Generalized Pustular Psoriasis

### 5.1. Introduction

GPP is a relatively rare variant of psoriasis, characterized by the appearance of erythematous plaques with neutrophilic, sterile pustular lesions, associated with systemic symptoms; the presence of fever, malaise, and elevation of inflammatory biomarkers is frequent and usually implies increases in morbidity and mortality [85].

The exact prevalence of GPP is unknown; however, it is considered a rare disease. It can occur in all races, and a certain preponderance in women has been reported. GPP presents as flares that require long-term control of the disease; these could be precipitated by external factors such as smoking, infections, pregnancy, and drugs. A paradoxical reaction to treatment with ustekinumab and TNF-α inhibitors has been described [86]. The low prevalence of this entity makes the diagnosis difficult. Likewise, there is scant evidence regarding the optimal management of this pathology [87].

### 5.2. Pathogenesis

The new advances in the pathophysiology of this disease have broadened the horizons of possible therapeutic targets for GPP:Mutations in IL36RN (gene that encodes the interleukin-36 receptor antagonist) have been identified in cases of GPP. These loss-of-function mutations result in the hyperactivation of IL-36 signaling. This induces neutrophil epidermal accumulation and the formation of pustules mediated by the production of inflammatory cytokines;Proinflammatory functions of IL-36 can be potentiated by a positive feedback loop with the IL-17/IL-23 axis. The sustained activation of IL-1 and IL-36 in GPP suggests that the IL-1/IL-36 inflammatory axis is the main physiopathological mechanism in GPP. IL-1 inhibition produces a partial response in patients with this entity, suggesting that IL-1 by itself may not play a central role in GPP; instead, IL-1 may act in a positive loop with IL-36;Mutations in caspase recruitment domain family member 14 (CARD14) and mutations in the adapter protein family 1 (AP1S3) have also been associated with pustular psoriasis [88].

### 5.3. Treatment

Treatment options include topical therapies, phototherapy, and systemic therapies.

#### 5.3.1. Phototherapy

Many phototherapy studies are case reports, and no randomized controlled trials (RCTs) have been conducted to date. They should be used with caution, and the dose should be adapted progressively, watching the skin for reactions after each irradiation. Any recommendations should be categorized as an expert opinion.

#### 5.3.2. Topical Therapies

Calcipotriene and tacrolimus have been used in monotherapies, and also combined with systemic therapies to treat severe disease [89].

#### 5.3.3. Retinoids

In a retrospective study that included 10 patients with GPP, acitretin resulted in a good but slow response, defined as the absence of new pustules within 3 days of treatment, and clearance of the majority of skin lesions within 4–6 weeks. However, it is important to note that a relapse could be observed upon acitretin withdrawal [90].

#### 5.3.4. Dapsone

Dapsone is not recommended for use as a first-line drug in cases of flares due to the slow onset of action. However, it could be considered as an alternative treatment when there is a poor response to first-line drugs. We initiated 50–100 mg of dapsone in two to three divided doses per day [85,89].

#### 5.3.5. Immunosuppressants


**Cyclosporine**


Because of its ability to yield immediate symptomatic relief, cyclosporine is usually considered as a first-line agent. The general approach is to initiate with cyclosporine 2.5–5.0 mg/kg per day (twice daily), and then adjust the dose based on symptoms [91].


**Methotrexate**


No RCTs have been performed, probably because of the small number of cases and the severity of disease, which makes large-scale comparisons difficult [92]. We recommend the use of methotrexate in patients with joint involvement, or in cases of refractoriness to the first-line treatment.


**Glucocorticoids**


Steroids by themselves could induce the formation of pustules, which is why we do not recommended them as a first-line therapy, although they could be of great help for use as an adjuvant therapy in cases of severe flares with systemic symptoms [89].


**Mycophenolate mofetil**


In a case study, improvements in skin lesions were described in one patient after seven days of treatment with mycophenolate mofetil; these were sustained for 4 months [93].

#### 5.3.6. Biological Treatment

##### TNF-α Inhibitors

Many case reports describe the efficacy of TNF-α inhibitors. However, a paradoxical reaction after administration has been reported. The physiopathology of this event is not fully understood. See Table 4. 


**Infliximab**


In a retrospective study, the administration of infliximab in a standard regimen (intravenous infusion at weeks 0, 2, and 6, and subsequently every 4 weeks) demonstrated flare control in four patients after 24–48 h of the infusion. Additionally, in an open-label study that included 10 patients with GPP flares, the time to pustular clearance ranged from 1 to 8 days [103].


**Adalimumab**


Adalimumab was described as effective and well tolerated for up to 52 weeks in 10 Japanese patients. Time to remission was variable, from 1 to 4 weeks [89].


**Etanercept**


A case series that included six patients with GPP showed a reduction in inflammatory biomarkers and clinical improvement [103].

##### IL-17A Inhibitors


**Brodalumab**


In an open-label, multicenter, long-term, phase III study, patients showed improved clinical status or remission after 12 weeks of treatment. By week 52, 91.7% were in clinical remission or had improved clinical status with the administration of brodalumab, an IL-17 receptor antagonist [96].


**Secukinumab**


Secukinumab is a monoclonal antibody that targets IL-17A, which demonstrated efficacy in a phase III study that included 12 patients with GPP in Japan. It was used as a monotherapy or in combination with other immunosuppressant drugs, and resulted in 9/12 patients (75%) achieving a Clinical Global Impression (CGI) score of “very much improved” at week 12, along with 7/12 patients (58.3%) achieving this at week 52 [89,94];


**Ixekizumab**


Another IL-17A antagonist, similar to secukinumab, is ixekizumab; efficacy has been demonstrated in patients with GPP in three phase III, open-label, multicenter studies, which included Japanese patients with GPP [89,104].

##### Anti IL-23 and IL-23/IL-12 Inhibitors


**Guselkumab**


Guselkumab is an IL-23 inhibitor that showed efficacy in a phase III, multicenter, open-label study in Japan that included 10 patients with GPP. In total, 50% of the patients showed improvement after one week of the administration of guselkumab [97].


**Risankizumab**


Risankizumab targets the p19 subunit of IL-23 and was approved in Japan for the treatment of patients with GPP [98,105].

##### Anti IL-1β and IL-1R Inhibitors


**Anakinra**


Anakinra is a recombinant IL-1 receptor antagonist, showing efficacy in reducing symptoms, normalizing inflammatory biomarkers, and stopping pustule formations in a patient with GPP and IL36RN mutation after 5 month of treatment [99].


**Canakinumab**


A monoclonal antibody that targets IL-1β, canakinumab, was used in a patient intolerant to anakinra, resulting in complete skin clearance and the improvement of systemic symptoms [100].


**Gevokizumab**


Gevokizumab blocks the activation of IL-1β receptors and has shown promising results in an open-label study in two patients with severe and refractory GPP [101].

##### IL-36 Pathway Inhibitor


**Spesolimab**


Spesolimab is a selective, humanized antibody against the IL-36 receptor that blocks its activation and suppresses the inflammatory response. In a phase II, multicenter, randomized, double-blind, placebo-controlled trial, patients treated with a single intravenous 10 mg/kg dose of spesolimab were more likely to achieve remission, defined by the clearance of pustular lesions within one week, than those in the placebo group [102]. 

### 5.4. Other Therapies

Therapeutic granulocyte and monocyte apheresis (GMA) is an extracorporeal circulation therapy that inhibits and removes neutrophils, macrophages, and monocytes, which accumulate in inflamed tissues. This therapy has shown efficacy in GPP only in case reports, case series, and reviews. No RCTs have been performed [89].

The majority of treatments discussed previously do not have a prescribing label for GPP, except in Japan, where TNF-α-blocking agents, IL-17/IL-17R inhibitors, and IL-23 inhibitors are approved for use against this disease. Brodalumab is also officially accepted in Taiwan and Thailand [96]. Recently, in September and October 2022, spesolimab was approved in Japan and the EU, respectively, for the treatment of acute symptoms in GPP [102]. Prospective studies are needed to review the outcomes and standardize the treatment of GPP.

## 6. Conclusions

In our review, we found that there are certain limitations when it comes to establishing action protocols in each disease, mainly due to the low quality of scientific evidence at present. Most of the publications are based on case reports. There are few randomized, controlled clinical trials; in most cases, the existing publications have small sample sizes, due to low prevalence rates. Likewise, in many cases, the drugs used for the treatment of neutrophilic dermatoses tend to have been developed recently, which is why we believe that it is essential to subsequently elucidate the pathophysiology and molecular pathways to extend the follow-up times of these patients and to closely monitor both the evolution and the side effects, in order to construct standardized treatment algorithms in the future.

## Figures and Tables

**Table 1 ijms-24-15622-t001:** Biological agents in the treatment of Sweet’s syndrome.

Biological Agent	Dosage	Reference
Adalimumab	40 mg every other week (alone or combined with systemic steroids)	[14]
Infliximab	5 mg/kg at weeks 0, 2, and 6, and every 6 to 8 weeks thereafter (alone or combined with topical or systemic steroids)	[11,12]
Ustekinumab	90 mg every 8 weeks	[13]
Anakinra	100 mg/day subcutaneously (combined with systemic steroids)	[15]
Rituximab	Rheumatoid arthritis protocol, (1000 mg at days 1 and 15) or 375 mg/m2 body surface, 2 additional cycles at 6 months and 18 months after the initial dose (combined with systemic steroids)	[14,16,17]

**Table 2 ijms-24-15622-t002:** Biological agents in the treatment of pyoderma gangrenosum.

Biological Agent	Dosage	Reference
Adalimumab	Induction dose of 160 mg at week 0, 80 mg at week 1, and 40 mg every 2 weeks.	[33,34,35]
Infliximab	5 mg/kg at weeks 0, 2, and 6, and every 6 to 8 weeks thereafter (alone or combined with topical or systemic steroids)	[29,30,31,32]
Certolizumab pegol	400 mg every other week for the first three injections, followed by 400 mg every 4 weeks (combined with systemic steroids)	[23,36]
Etanercept	50 mg every 2 weeks	[32]
Golimumab	200 mg at week 0, 100 mg at week 2, and every 4 weeks thereafter (combined with topical or systemic steroids)	[37]
Ustekinumab	90 mg every 8 weeks or at week 0, week 4, week 8, and every 8 weeks for a minimum of 12 months. Usually administered in the absence of concomitant therapy	[38,39,40,41]
Secukinumab	300 mg subcutaneously initially and later at weeks 0, 1, 2, 3, and 4, followed by a monthly maintenance dose	[42,43]
Canakinumab	150 mg of canakimumab subcutaneously at week 0, with a possible extra dose after two weeks in cases of refractoriness, as well as an optional dose of 150–300mg at week 8	[44]
Anakinra	100 mg daily subcutaneously, in combination or not with glucocorticoids, which can be increased to 200 mg daily (2mg/kg day) in cases of inadequate clinical response, or 4 weeks of loading at 2 mg/kg daily, followed by 100 mg once daily	[45,46,47,48]

**Table 3 ijms-24-15622-t003:** Biological agents in the treatment of Hidradenitis suppurativa.

Biological Agent	Dosage	Reference
Adalimumab	80 mg s.c. first week, 40 mg s.c. second week	[68,71]
Infliximab	5 mg/kg body weight is administered on week 0, 2, 6, and then regularly every 8 weeks	[68]
Anakinra	100 mg/day subcutaneously	[72,73,74]
Ustekinumab	45 mg s.c. at weeks 0, 4, 16, and 28	[68,75]
Secukinumab	300 mg s.c. every 2 weeks, every 4 weeks	[77,78]

**Table 4 ijms-24-15622-t004:** Biological agents in the treatment of generalized pustular psoriasis.

Biological Agent	Dosage	Reference
Etanercept	50 mg of etanercept twice weekly subcutaneously	[94]
Infliximab	5 mg/kg body weight is administered on week 0, 2, 6, and then regularly every 8 weeks	[95]
Adalimumab	Dose of 80 mg is given s.c. in adults, and after the second week, a dose of 40 mg is given s.c. every 2 weeks. If the efficacy is insufficient, the dose may be increased to 80 mg/administration	[89]
Brodalumab	s.c. injections of 210 mg at weeks 0, 1, 2 and then every 2 weeks thereafter	[89,96]
Secukinumab	Weekly s.c. injections of 300 mg on weeks 0, 1, 2, 3, and 4, and then every 4 weeks thereafter (may be decreased to 150 mg)	[89]
Ixekizumab	s.c. injections of 160 mg at week 0, followed by 80 mg at weeks 2, 4, 6, 8, 10, and 12, then 80 mg every 4 weeks thereafter	[89]
Guselkumab	100 mg dose subcutaneously on weeks 0 and 4, followed by 100 mg every 8 weeks	[97]
Risankizumab	150 mg subcutaneously plus two additional samples at weeks 4 and 16	[98]
Anakinra	100 mg daily subcutaneously	[99,100]
Canakinumab	150 mg subcutaneously per month	[100]
Gevokizumab	60 mg subcutaneously every 4 weeks for a total of three injections (12 weeks)	[101]
Spesolimab	Single intravenous dose of 10 mg/kg	[102]

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
