# Peer review of "Treatment Strategies in Neutrophilic Dermatoses: A Comprehensive Review"

_ijms, 2023, doi:10.3390/ijms242115622_

Round 1

Reviewer 1 Report

To the Authors:

I have read your review article regarding the treatment protocols for neutrophilic dermatosis and other autoinflammatory diseases. While the review provided valuable data, I would like to address a few major issues:

1. The first part of the title is somewhat unclear: "New dermatological treatments in a complex patient." Authors should provide more specific information or clarify the complexity being discussed.

2. The focus on treatment protocols for neutrophilic dermatosis autoinflammatory patients is commendable. However, it is necessary also to describe the shared possible immunopathological mechanisms, as understanding these mechanisms is critically important in this context.

3. In light of the title and study goal, which aimed to investigate new treatment protocols shared among neutrophilic dermatosis and autoinflammatory patients, the conclusion stating that "it is necessary to extend the follow-up times of these patients" seems to raise a fundamental question. Why did the authors choose to investigate this subject if there were limitations that prevented any conclusive findings? It seems that authors should completely reconsider the aim of the study.

I hope these points provide constructive feedback and help enhance the overall clarity and impact of your review.

Author Response

Thank you very much for taking the time to review this manuscript. Please find the detailed responses below and the corresponding revisions/corrections highlighted in the re-submitted file. 

1- Thank you for pointing this out, we have adjusted the title according to the given recommendations.

2-  We have, accordingly, modified the structure to emphasize on the "etiopathogenesis" section. We separate the information to provide a brief summary before each pathology, as it is not the objective of the study.

3- Agree, this would probable be a good opportunity to explore our posibilities of making our own studies that could provide scentific evidence.  For now, we have only done a descriptive review, trying to summarize and standardize information. 

Reviewer 2 Report

The paper presents an interesting and comprehensive review on neutrophilic dermatosis.

However, the title of the manuscript is missleading because the text is not directly focused on "a complex patient" but it is rather the description of the selected diseases together with the disscusion on various methods of treatment. Also, there are many other autoinflammatory diseases which are not mentioned in the paper. I suggest authors to correct the title before publication.

My next remark is that the text is in some points chaotic. While describing the diseases the authors should organize the text in the same way. Please, include similar subpoints while describing each disease. E.g. in the part on Sweet syndrome there are no subpoints whereas in the part on pyoderma gangrenosum they have included subpoints (3.1., 3.2., etc.), in part on Hidradenitis suppurativa and PPP the authors have used dots.

Author Response

Thank you very much for taking the time to review this manuscript. Please find the detailed responses below and the corresponding revisions/corrections highlighted in the re-submitted file. 

1- Thank you for pointing this out, we have adjusted the title according to the given recommendations.

2-  We have, accordingly, changed and re-structured the article including similar subpoints while describing each disease. 

Round 2

Reviewer 1 Report

Thank you for considering the issues. Your manuscript is almost complete